# Characterization of *Bifidobacterium asteroides* Isolates

**DOI:** 10.3390/microorganisms10030655

**Published:** 2022-03-18

**Authors:** Alessandra Pino, Bachir Benkaddour, Rosanna Inturri, Pietro Amico, Susanna C. Vaccaro, Nunziatina Russo, Amanda Vaccalluzzo, Gianluigi Agolino, Cinzia Caggia, Hadadji Miloud, Cinzia L. Randazzo

**Affiliations:** 1Department of Agricultural, Food and Environment, University of Catania, 95123 Catania, Italy; alessandra.pino@unict.it (A.P.); nunziatina.russo@unict.it (N.R.); amanda.vaccalluzzo@unict.it (A.V.); gianluigi.agolino@unict.it (G.A.); ccaggia@unict.it (C.C.); 2ProBioEtna S.r.l., Spin-Off of University of Catania, 95123 Catania, Italy; 3Department of Biology, Faculty of Natural Sciences and Life, University of Oran1, Oran 31000, Algeria; benkaddourbachir@gmail.com (B.B.); hadadjimiloud@yahoo.fr (H.M.); 4Department of Biomedical and Biotechnological Sciences, University of Catania, 95123 Catania, Italy; 5Department of R&D, Local Noto Unit, Fidia Farmaceutici S.p.A., 96017 Noto, Italy; pamico@fidiapharma.it (P.A.); svaccaro@fidiapharma.it (S.C.V.)

**Keywords:** microbiological characterization, safety, *vanZ*, isolation, vancomycin resistant gene, genome, bee, honey

## Abstract

Bifidobacteria have long been recognized as bacteria with probiotic and therapeutic features. The aim of this work is to characterize the *Bifidobacterium asteroides* BA15 and BA17 strains, isolated from honeybee gut, to evaluate its safety for human use. An in-depth assessment was carried out on safety properties (antibiotic resistance profiling, β-hemolytic, DNase and gelatinase activities and virulence factor presence) and other properties (antimicrobial activity, auto-aggregation, co-aggregation and hydrophobicity). Based on phenotypic and genotypic characterization, both strains satisfied all the safety requirements. More specifically, genome analysis showed the absence of genes encoding for glycopeptide (*vanA*, *vanB*, *vanC*-1, *vanC*-2, *vanD*, *vanE*, *vanG*), resistance to tetracycline (*tetM*, *tetL* and *tetO*) and virulence genes (*asa1*, *gelE*, *cylA*, *esp*, *hyl*).

## 1. Introduction

Bifidobacteria are Gram-positive, non-motile and non-spore-forming bacteria with a curved and clubbed shape, often branched with Y and V forms. Their genome harbors a high G + C content. Increased interest in LAB and bifidobacteria has been registered in the last two decades [1]. Bifidobacteria are commonly isolated from the gastrointestinal tract of various animals such as mammals, birds and insects. However, strains ascribed to *Bifidobacterium denticolens*, *Bifidobacterium dentium*, *Bifidobacterium inopinatum* and *Bifidobacterium animalis* subsp. *lactis* species have been isolated from different ecological niches, such as the human oral cavity [2,3,4] and fermented milk products [5,6]. *Bifidobacterium asteroides*, *Bifidobacterium coryneforme* and *Bifidobacterium indicum* are species that were isolated and characterized in the 1960s in pollinating insects, including honeybees *Apis mellifera*, *Apis cerana* and *Apis dorsata* [2,7]. In the last decades, the new species of *Bifidobacterium bombi*, *Bifidobacterium actinocoloniiforme*, *Bifidobacterium bohemicum* and *Bifidobacterium commune* have been identified from *Bombus* spp. gut [8]. While gut microbiota in some insects is acquired from food and the environment, bifidobacterial populations in honey, wasps and bumble bees are stable, and different molecules have been characterized as mediators of their *cross-talk* with the host [8]. In fact, similarly to humans, honeybees have a core in their gut microbiota that matures in the early stages of their life and remains stable throughout adult life. The gut of mammals, birds and insects is known as an anaerobic environment, and bifidobacterial growth has been described as being inhibited by oxygen, which exerts toxic effects. However, the *Bifidobacterium asteroides* species has been found to tolerate oxygen levels higher than 20%, which represents the upper limit of tolerance for *Bifidobacterium animalis* subsp. *lactis*, *Bifidobacterium boum*, *Bifidobacterium psychraerophilum* and *Bifidobacterium thermophilum* [7]. While the long history of safe use of some species, such as *Bifidobacterium adolescentis*, *Bifidobacterium animalis*, *Bifidobacterium longum*, *Bifidobacterium breve* and *Bifidobacterium bifidum*, has led to them being proposed for European Safety Authority (EFSA) Qualified Presumption of Safety (QPS) status, other species, such as *Bifidobacterium dentium*, cannot be used because they are linked to cariogenic processes. Moreover, the current guidelines do not authorize the use in humans of strains of non-human origin [9].

Recently, researchers have turned their attention to different species of honeybees including *Apis mellifera*, on account of their elevated mortality rates. Honeybees and other pollinating species play a key role in production and global food security, which amounts to an estimated value of 150 billion euro. Many funding programs for pollinator insects have set up grants for the protection of specific species and their habitats. Decades-long investigations on insect physiology have made it possible to understand the single and combined effects of environmental and host stress factors such as pests, pathogens, toxins and nutrient-limited food sources. Moreover, even if the vast majority of probiotic strains have been isolated from human or fermented foods, mainly dairy, the scientific research has been recently focused on the selection of new probiotics from *unconventional* sources such as meat, fruit, vegetables, cereals, honeybees and beehive products. In addition, the Internal Scientific Association for Probiotics and Prebiotics (ISAPP) recognized probiotic properties of strains belonging to *Akkermansia muciniphila*, *Faecalibacterium prausnitzii*, *Roseburia* spp. and *Eubacterium hallii*, not previously included in the QPS list [10,11,12,13,14].

Nowadays, studies on the gut microbiota of pollinators have clarified the role of the genera *Bifidobacterium* and *Lactobacillus* in immunity preservation, disease tolerance and resistance to environmental stressors [15]. The core of honey bees’ gut microbiota, whatever their geographical origin, is composed of *Gilliamella apicola*, *Lactobacillus Firm-5* (*L. melliventris*, *L. kimbladii*, *L. kullabergensis*, *L. helsinborgensis*), *Lactobacillus Firm*-4 (*L. mellis*, *L. mellifer*), *Snodgrassella alvi* and *Bifidobacterium asteroides* [16]. The gut microbiota and its symbiotic bacteria exert a key role in the innate immunity system of bees, despite the production of antimicrobial peptides with a highly selective activity against pathogenic species [15]. The disruption of the microbial balance in the bee’s gut is due to the perturbation of insect immunity through treatments with microbicidal or microbiostatic compounds; the exposure to pesticides or herbicides at a sublethal concentration (called hidden treatment) leads to a decrease in *Firmicutes* and *Actinobacteria* and to an increase in *Gammaproteobacteria*, such as *Gilliamella apicola* and *Escherichia coli*. Gut microbiota dysbiosis and the dysregulation of the innate immunity system expose bees to attacks by parasites such as fungi (*Nosema* spp.) and trypanosomes (*Crithidia* and *Lotmaria* spp.). Moreover, honeybee gut plays a key role in their metabolic activity and nutrition status, especially impacting on vitamin biosynthesis.

Based on knowledge of human gut dysbiosis, two strategies have recently been used to control honeybee dysbiosis and dysregulation: a prebiotic strategy through diet supplementation with sucrose-based solutions or pollen-based feed, and gut microbiota manipulation through supplementation with safe strains from the *Bifidobacterium* and *Lactobacillus* genera [16,17].

There is a lack of studies in the literature reporting the selection of *Bifidobacterium* strains as feed supplementation for honeybees. However, several regulatory authorities have suggested that these strains could possess some important properties that make them suitable for human use. Studying the resistance genes is important for the confirmation of the isolates as non-resistant bacteria. Antibiotic resistance can be due to phenotypic or genotypic features, and the antibiotic resistance profiles of the *Bifidobacterium asteroides* species could be the result of long-term exposure of honeybees to antibiotics [18]. Recent studies have indicated horizontal antibiotic resistant gene transfer between bacteria residing in the guts of humans and animals. These genes are delivered by mobile genetic elements; the collection of the mobile elements of a microorganism has recently been defined as a *mobilome* and includes transposases, insertion elements, plasmids and prophages (bifidoprophages). The entire *mobilome* and *resistome* have recently been reconstructed. The most abundant antibiotic resistant gene in the *Bifidobacterium resistome* is the one conferring glycopeptide resistance (43%, 5999 putative enzymes), followed by the methyltransferase class (19%, 2618 putative enzymes), β-lactamase class (17%, 2437 putative enzymes), tetracycline class (16%, 2178 putative enzymes), sulfonamide class (4%) and metronidazole and aminoglycosides classes (0.5%). Analysis of the putative mobile *resistome* of the *Bifidobacterium* genus reveals the presence of: hypothetic conjugative transposon that harbors the *tetW* gene, responsible for protection from tetracycline activity; putative prophage-like elements, which harbor the *BacA* gene responsible for protection from bacitracin; transposase that harbors a 23S rRNA methylase, which confers resistance towards erythromycin and clindamycin. Moreover, among the predicted transposase encoding genes, the presence of the *vanZ* gene has been observed, which may confer low-level resistance to glycopeptide antibiotics [19].

One of the main selection criteria for defining the probiotic action of bifidobacterial and lactic acid bacteria is adhesion to human intestinal cells. Adherent strains could exert metabolic and immunomodulatory functions, stabilizing the intestinal mucus barrier and providing the competitive exclusion of pathogenic bacteria [20,21,22,23,24]. Exopolysaccharides can be involved in the adhesion to mucus [25]. The presence of genes and gene clusters encoding for *pilus-like* structures has also recently been demonstrated in bifidobacterial genomes [26]. The degree of in vitro adhesion depends on many factors, such as the substrate used for the assay (abiotic or biotic surface) and growth medium composition.

*Bifidobacterium asteroides* has peculiar properties, such as the ability to tolerate oxygen and other metabolic features for carbohydrate metabolism, which have not been already determined in the genus of *Bifidobacterium*. Genome analysis of the strain *Bifidobacterium asteroides* PLR2011species demonstrated the presence of a “malolactate fermentation pathway”, responsible for the conversion of malic acid to lactate, with the addition of the characteristic “fructose-6-phosphate pathway”. The latter is uniquely responsible for fructose and glucose fermentation and contains fructose-6-phosphate phosphoketolase (F6PPK), but glucose-6-phosphate is absent [7] Moreover, the *Bifidobacterium asteroides* species is reported to possess catalase [27].

In the present study, two *Bifidobacterium* strains isolated from honeybee gut were evaluated as to their safety for human use and other properties.

## 2. Materials and Methods

### 2.1. Reference Strains and Culture Conditions

The hemolytic Gram-positive strains *Streptococcus pyogenes* ATCC 19615 and *Streptococcus pneumoniae* ATCC 6303 were cultured on Brain Heart Infusion (BHI, Becton Dickinson GmbH, Heidelberg, Germany) at 37 °C under 5% CO_2_ conditions. *Escherichia coli* ATCC 25922, *E. coli* ATCC 9637, *Staphylococcus aureus* ATCC 6538 and ATCC 29213 were routinely cultured on Trypticase Soy Broth medium (Oxoid, Milan, Italy) at 37 °C under aerobic conditions. *Listeria monocytogenes* DSM 12464, *Salmonella enterica* serovar *typhimurium* ATCC 14028 and *Salmonella enterica* subsp. *enterica* serovar *enteritidis* ATCC 13076 were revitalized in BHI broth at 30 °C under aerobic conditions. The probiotic strain *Bifidobacterium animalis* BB12 (Christian Hansen AS, Hoersholm, Denmark), was grown in Bifidus Selective Medium Broth (BSM, Sigma Aldrich, Milan, Italy) or in Man Rogosa and Sharpe (MRS, Oxoid, Milan, Italy) supplemented with 0.25% L-cysteine (Sigma Aldrich, Milan, Italy) (MRSc) at 37 °C, under anaerobic conditions, and included in every experiment for comparison.

### 2.2. Identification of Isolates

Two bacterial isolates named BA15 and BA17, previously obtained from honeybee gut and subjected to morphological characterization (unpublished data), were analyzed for physiological and biochemical properties (catalase, oxidase, spore formation, gelatinase activities, production of indole, NH_3_ from arginine and CO_2_ from glucose) and with the use of the API rapid ID 32 A kit (BioMérieux, Grassina, Italy). Based on the enzymatic profile, typical of the *Bifidobacterium* genus, both isolates were identified at species level. The total genomic DNA was extracted following the method previously described [28] and 16S rDNA was amplified using the primer pairs Bif164 and Bif662 [29,30]. PCR products were purified using a Qiaquick PCR purification kit (Qiagen Hilden, Germany) and subjected to 16S rDNA sequencing. Comparison with sequences held in the BLAST database allowed both stains to be ascribed to the *Bifidobacterium asteroides* species. The accession numbers of the sequenced strains were as follows (code and identity percentage of isolates in parentheses): *Bifidobacterium asteroides* MG650026.1 (BA15, 99.60%) and *Bifidobacterium asteroides* CP017696.1 (BA17, 99.41%).

### 2.3. Safety Assessment

#### 2.3.1. Antimicrobial Susceptibility and MIC Determination

Antimicrobial susceptibility was determined according to the ISO 10932:2010 [31] broth microdilution procedure using eight antimicrobial agents (ampicillin sodium salt, chloramphenicol, clindamycin hydrochloride, erythromycin, gentamicin sulphate, streptomycin sulphate salt, tetracycline, vancomycin), all purchased from Sigma Aldrich (St. Louis, MO, USA). The Minimum Inhibitory Concentration (MIC), defined as the lowest concentration of antibiotic giving a complete inhibition of visible growth in comparison to an antibiotic-free control well, was determined by the microdilution method according to Russo et al. (2018) [32]. The experiments were conducted in triplicate.

The genomes of the BA15 and BA17 strains were analyzed for the presence of antibacterial resistant genes and other gene associations that can influence the safety profile of the strains. The analysis was performed using the Pathosystems Resource Integration Center (PATRIC) database [33].

#### 2.3.2. PCR Assay on Virulence Factors

The presence of virulence genes (*asa1*, *cylA*, *gelE*, *hyl*, *esp*) was evaluated by multiplex PCR following the method described by Vankerckhoven et al. (2004) [34] and using the primer pairs reported in Table 1. PCR reactions were performed in a final volume of 25 µL containing 1.0 µL of genomic DNA, 1.0 µL of each primer (100 mM) and 12.5 µL of 5-PRIME MasterMix including HotStarTaq DNA polymerase (Eppendorf, Milan, Italy). Amplification was carried out as follows: an initial activation step at 94 °C for 15 min, where DNA polymerase was activated; 30 amplification cycles of denaturation (94 °C for 1 min), annealing (56 °C for 1 min) and extension (72 °C for 1 min); followed by one final extension step consisting of 10 min at 72 °C. PCR products were analyzed by electrophoresis in a 1.5% *w*/*v* of agarose gel for 1 h at 90 V in 1.0 × TAE buffer solution. After treatment with ethidium bromide solution, the amplicons were detected by UV light. The Lamba DNA/HindIII marker (Thermo Fisher Scientific, Rodano, Italy) and the Φ174 DNA Marker Hae III Digest (Sigma Aldrich, Milan, Italy) were used as DNA ladders [35].

#### 2.3.3. Other Assessments: Hemolytic Activity and DNase and Gelatinase Activities

*B. asteroides* BA15 and BA17 strains, grown in BSM broth for 18–24 h at 37 °C under anaerobic conditions, were streaked onto blood agar plates containing sheep blood (Biolife, Milan, Italy), and anaerobically incubated at 37 °C for 24–48 h. After incubation, the plates were visually analyzed for the presence or absence of microbial hemolytic properties and distinguished as β-hemolysis, α-hemolysis or γ-hemolysis, based on the appearance of a clear zone, green halo or no zones around the colonies, respectively [28]. *S. pyogenes* ATCC 19615 and *S. pneumoniae* ATCC 6303 were used as positive controls. DNase and gelatinase activities were tested in triplicate, as suggested by Pino et al. (2019) [36]. In detail, to evaluate the DNase activity, 5 μL of a liquid culture was spotted onto DNase agar plates (Oxoid, Milan, Italy). After incubation at 37 °C for 48 h, plates were overlain with HCl 1 N for 5 min. Gelatinase activity was assessed by spotting 5 μL of an overnight culture of the strain on BHA supplemented with 0.04% gelatine.

### 2.4. Hydrophobicity, Auto-Aggregation and Co-Aggregation Abilities

Hydrophobicity (H%), auto-aggregation (Auto-A%) and co-aggregation (Co%) abilities were tested as described by Pino et al. (2021) [37]. Hydrophobicity was determined as bacterial adhesion to hydrocarbons (BATH) using xylene. *E. coli* ATCC 25922, *S. aureus* ATCC 6538 and *S. typhimurium* ATCC 14028 were used as the pathogenic strains in the co-aggregation assay.

### 2.5. Adhesion on Abiotic Surface

Adhesion capability was tested using overnight cell cultures grown in Man Rogosa and Sharpe (MRS, Oxoid, Milan, Italy) broth supplemented with 0.25% L-cysteine (Sigma Aldrich, Milan, Italy) (MRSc) and incubated at 37 °C under anaerobic conditions (80% N_2_, 10% CO_2_ and 10% H_2_) using the AnaeroGen sachet (Oxoid, Milan, Italy). Then, 200 μL of a 1:100 dilution of each culture was transferred to a 96-well micro-ELISA plate (number of replicates: 32) and, after regular shaking, the absorbance at t_0_ (starting time) was read at 600 nm (ELx808, BioTek-software Gen5) and the plate was incubated. After incubation at 37 °C for 120 min under aerobic conditions, the plate was washed twice using sterile PBS to remove non-adherent bacteria and air-dried for 60 min at 60 °C. Then, 200 μL of a solution of 0.25% crystal violet was added to each well and the plate was incubated at room temperature for 15 min. After incubation, the plate was rinsed twice using Milli-Q water (Millipore, Milan, Italy) to remove excess dye and 200 μL of a 98% ethanol solution was added to each well. The absorbance was read at 570 nm. The adherence index was calculated as follows: Abs570/Abs600 [38].

### 2.6. Alignment for VanZ Putative Gene

The putative *VanZ* gene sequences for the *Bifidobacterium asteroides* BA15 and *Bifidobacterium asteroides* BA17 genomes were compared to those of reference strains *Bifidobacterium asteroides* DSM 20089 (CP017696.1:1959032-1960146) and *Bifidobacterium asteroides* PRL2011 (CP003325.1:10964-12078) using a multiple sequence alignment method with reduced time and space complexity (MUSCLE) and DNASTAR software [39].

### 2.7. Antagonistic Activity against Pathogens

#### 2.7.1. Agar Diffusion Assay

Antagonistic activity was evaluated using *E. coli* ATCC 25922, *E. coli* ATCC 9637, *S. aureus* ATCC 6538, *S. aureus* ATCC 29213 and *S. typhimurium* ATCC 14028 as target bacteria. The assay was performed by the agar spot test [40], using the cell-free culture supernatants obtained as reported by Argyri et al. (2013) [41]. After incubation for 48 h, the appearance of inhibition zones was visually detected and, based on the diameter size, results were expressed as: (−) no inhibition zone; (+) inhibition zone < 5 mm; (++) inhibition zone > 5 mm [21].

#### 2.7.2. Antibacterial Activity

Antagonistic activity was evaluated using *E. coli* ATCC 9637 and *S. aureus* ATCC as models for Gram-negative and Gram-positive bacteria, respectively. The assay was performed modifying the broth microdilution method described by CLSI M7-A7 for bacteria [24,42], using the cell-free supernatants obtained as previously described. Incubation was performed under aerobic conditions at 37 °C for 24 h. The absorbance was read at 630 nm (ELx808, BioTek-software Gen5) after regular shaking with a frequency every 30 min. The killing curves were created by plotting OD values versus time, and bacterial growth kinetics were studied using GraphPad Prism 8 software (Version 8.2.1_279). Each assay was performed three times in duplicate.

### 2.8. Statistical Analysis

All data were expressed as a mean and standard deviation of three independent experiments. Data were subjected to one-way ANOVA followed by Tukey’s Multiple Comparison Test and differences were considered statistically significant at α = 0.05.

## 3. Results

### 3.1. Enzymatic Profile

Table 2 shows the enzymatic profile of the tested *B. asteroides* BA15 and BA17 strains as well as the percentage of the positive reactions for the *Bifidobacterium* genus, carried out according to the manufacturer’s instructions. The results of the assay, analyzed by apiweb™ suggested an enzymatic profile for both strains that was characteristic of the *Bifidobacterium* genus (% ID 99.9). More precisely, the ID profile was 4537033705 for the BA15 strain, whereas the ID profile for the BA17 strain was 4517033505 (Table 2). Based on the fermentative profile (Table 3), both strains showed similar biochemical properties except for their ability to ferment L-Arabinose.

### 3.2. Antibiotic Resistance Profile, Virulence Factors and Other Biochemical Properties

The antibiotic susceptibility profile for both *B. asteroides* BA15 and *B. asteroides* BA17 strains, based on EFSA criteria [43], showed susceptibility to the main tested antimicrobials, except for ampicillin, vancomycin and chloramphenicol (Table 4).

The PCR-based approach did not reveal the presence of genes encoding for gelatinase (*gelE*), hyaluronidase (*hyl*), aggregation substance (*asa1*), enterococcal surface protein (*esp*) and cytolysin (*cylA*).

The results obtained from the analysis of the bacterial genome using PATRIC bioinformatic services showed the absence of antibiotic resistant genes in the BA15 and BA17 genomes and the presence of a metabolic pathway responsible for vancomycin biosynthesis (Figure 1). The key enzyme was the dTDP-glucose 4,6-dehydratase (EC number 4.2.1.46), encoded by a gene located from nucleotide 1368973 to 1370001 for the BA15 strain, and from nucleotide 319105 to 320202 for the BA17 strain. The expression of this enzyme and production of vancomycin needs further investigation; however, this metabolic pathway could explain the resistance to vancomycin.

None of the tested *Bifidobacterium asteroides* strains showed the ability to produce DNase and gelatinase or to exert hemolytic activity.

Antagonistic activity against food spoilage and pathogenic bacteria shown by the BA15 and BA17 strains is reported in Table 5. Overall, both BA15 and BA17 strains showed antagonistic activity against all tested pathogens, with the exception of the BA15 strain, which did not show any antagonistic activity against *S. aureus* ATCC 6538 (Table 5).

### 3.3. Adhesion to Abiotic Surfaces

Figure 2 shows the ability of *B. asteroides* BA15 and BA17 strains to adhere on abiotic surfaces (expressed as *adherence index*). Overall, both tested strains showed adhesion abilities and the highest adherence index was exhibited by the BA17 strain (4.00).

### 3.4. Alignment for VanZ Putative Gene

The *vanZ* gene is an orthologous gene belonging to the glycopeptide resistance protein family (*vanZ*-A, *vanZ*-F, *vanZ*-Pt and *vanZ*-1). Both tested stains showed the presence of a putative *vanZ* gene. More specifically, *vanZ* was detected in the genome of the *B. asteroides* BA15 strain in three different positions (Strain15_0001: 457-972; Strain15_0979: 1060104-1060616; Strain15_2441: 2576152-2577315), whereas it was only identified once in the *B. asteroides* BA17 genome (17_1679: 2161147-2162262). On the basis of *vanZ* nucleotide analysis using blastn, the *Strain15_0001*, *Strain15_0979* and *Strain15_2441* genes showed a high (100% identity) homology with the genes present in the *Lactobacillus plantarum* genomes. The gene 17_1679 showed a high (93.9% identity) homology with the gene present in the *Bifidobacterium asteroides* genomes.

Figure 3 shows the putative proteins encoded by *vanZ* from *B. asteroides* BA15 and *B. asteroides* BA17. The number of amino acids of the putative protein encoded by the *vanZ* gene results as being 171 for gene *Strain15_0001*, 170 for gene *Strain15_0979*, 387 for gene *Strain15_2441* and 371 for gene *Strain17_1679*, which is equal to those for the control strain PRL 2011. All proteins showed the conserved domain *vanZ*, which belongs to the *vanZ*-like family and contains several examples of the *vanZ* protein, as well as examples of phosphotransbutyrylases; however, they differ in amino acid length. Moreover, the putative protein encoded by *vanZ* from gene *Strain15_2441* also showed the RDD domain, which is a family of proteins that contains three highly conserved amino acids (one arginine and two aspartates). This region contains two predicted transmembrane regions: the arginine occurs at the N-terminus of the first helix and the first aspartate occurs in the middle of this helix. The molecular function of the RDD region is unknown; however, this region may be involved in the transport of a set of ligands that are still not well identified.

Table 6 shows the ratio between percentage identity and distance of the *vanZ* genes of BA15 and BA17 in comparison with the genes present in the genome of the reference strain PRL2011. The MUSCLE analysis algorithm showed a high homology (94%) between the putative *vanZ* gene from *B. asteroides* BA17 and the reference strain *B. asteroides* PRL2011 (CP003325.1:10964-12078). Instead, a lower homology percentage (from 46.5% to 49.8%) was shown by *VanZ* genes from the *Bifidobacterium asteroides* BA15 genome (Table 6).

Figure 4 shows the unrooted phylogenetic tree relating the *vanZ* genes of BA15 and BA17 strains and their distance with the PRL2011 strains. Moreover, the nucleotide sequences of the putative *vanZ* gene of BA15 and BA17 strains, aligned using MUSCLE (MegAlign Pro of DNAstar), compared with DSM 20089 and PRL2011 strains, are shown in the Appendix A.

### 3.5. Antibacterial Activity

Figure 5 shows the antibacterial effect of the BA15 and BA17 strains against *S. aureus* ATCC 29213 and *E. coli* ATCC 9637. Both tested strains showed high activity against the Gram-positive *S. aureus* ATCC 29213, with a dilution factor of wild supernatant from 1:4 to 1:32 ratio (Figure 5A,C). Both tested supernatants resulted as being active towards the Gram-negative *E. coli* ATCC 9637 in a dilution range from 1:4 to 1:8 (Figure 5B,D). The antibacterial activity against *S. aureus* ATCC 29213 was exerted in the dilution range from 1:4 to 1:8, whereas, against *E. coli* ATCC 9637, it was only exerted at the higher tested concentration (dilution 1:4 of the wild supernatants). The supernatant obtained from BA15 and BA17 showed an inhibitory activity towards *S. aureus* ATCC 29213 when it was tested at the dilution range from 1:16 to 1:32, and towards *E. coli* ATCC 9637 when it was tested at a dilution of 1:8. In particular, both BA15 and BA17 appeared to reduce the growth of *S. aureus* ATCC 29213 and *E. coli* ATCC 9637, which after 24 h of incubation reported an OD_630_ lower than the untreated control. The growth of *E. coli* ATCC 9637 was delayed by about 14 h from the time of incubation when treated with BA15 and by about 9 h (from the time of incubation) when treated with BA17, although after 24 h, the OD_630_ values were the same as those of the untreated control.

### 3.6. Auto-Aggregation, Co-Aggregation and Hydrophobicity Abilities

Table 7 summarizes the surface characteristics (hydrophobicity, auto-aggregation and co-aggregation) of the *B. asteroides* BA15 and BA17 tested strains, compared to those exhibited by the *Bifidobacterium animalis* BB12 strain. The BA15 and BA17 strains showed an auto-aggregation percentage, similar to that displayed by the BB12 reference strain. Both BA15 and BA17 strains also exhibited the ability to co-aggregate with the tested pathogens. In particular, both the *B. asteroides* BA15 and BA17 strains showed the highest co-aggregation percentage with *Listeria monocytogenes* ATCC 12466 (Table 7) when compared to the *B. animalis* BB12 strain. A variable degree of hydrophobicity was observed, with the highest percentage exhibited by the BB12 and BA17 strains.

## 4. Discussion

The gut microbiota of honeybees and insect pollinators is still an unexplored ecosystem. Recent findings on human gut microbiota have paved the way for a better understanding of its role in other living species on Earth. The evolutionary role of bifidobacteria among humans and animals seems to be related to their ability to ferment complex non-digestible carbohydrates and to modulate the host immune system through changes in innate and/or adaptive immune responses [44,45]. The distribution of specific species of bifidobacteria across the human lifetime has been recently studied [46] and a high transfer level of the species belonging to the main phyla of gut microbiota between host family members has been demonstrated [46,47]. A small number of bifidobacterial subspecies (*Bifidobacterium pseudolongum*, *Bifidobacterium adolescentis*, *Bifidobacterium pseudocatenulatum* and *Bifidobacterium bifidum*) have been recognized as cosmopolitan because they have been isolated from various animal and mammalian hosts, unlike other taxa which appear to be much less widely distributed [47]. A study of primate-associated bifidobacteria demonstrated the phylosymbiosis between the Hominidae family and bifidobacterial species isolated from humans, on the basis of observed bifidobacterial–host co-phylogeny [48]. Despite the social relevance of bees, their gut microbiota is still far from being completely understood. Recent ecological surveys on gut microbiota of insects have revealed that in the same way as mammals, they rely on a mutualistic gut microbial community [49]. Differently from other insects, such as ants, whose microbiota is acquired from food and the environment, honeybees, similarly to humans, have a gut microbiota with a stable core that, after the early developmental stages, remains relatively stable through most of their adult lifetime [50]. It has recently been discovered that the phyla that constitute the core of honeybee gut microbiota are three of the most important components of human gut microbiota (Firmicutes, Proteobacteria, Actinobacteria) [15]. Several studies using 16S rDNA surveys and metagenomic of the total DNA, highlighted that *Bifidobacterium asteroides*, along with *Lactobacillus* FIRM4 and *Lactobacillus* FIRM5, represent the so-called *core-bacteria*, as the most essential microorganism in the honeybee gut and these evidence could be related to a possible probiotic potential of *Bifidobacterium asteroides* strains [49,50,51,52]. The *Bifidobacterium asteroides* PRL2011 strain, isolated from the hindgut of *Apis mellifera*, represents the first reported case of the presence of a respiratory chain, which means that this strain may be able to grow in aerobic conditions. This species is phylogenetically distant from other bifidobacterial species, and its ability to tolerate oxygen has been lost in bifidobacteria that inhabit the mammalian gut [7,51,52]. The present study characterizes two strains of *Bifidobacterium asteroides* isolated from honeybees, with the aim of contributing to a better understanding of the properties of this species and their possible applications.

The strains were typed and characterized by using both phenotypic and genotypic tests, following the FAO/WHO working group [53] guidelines, to identify the strains at phenotypic/genotypic levels. In addition, stepwise in vitro procedures were carried out to investigate safety and other properties. The EFSA has suggested that more research on bacterial genomes should be carried out to provide an adequate characterization of new isolates [54]. Genome analysis can define the safety profile and is useful for the characterization of specific properties, such as the production of metabolites, polysaccharides and compounds with antimicrobial activity [7,55,56,57].

In accordance with former studies, our results confirm that the combination of genotypic and phenotypic methods is a powerful tool for strain discrimination [30,58].

Safety concerns represent one of the main requirements that should be addressed for the selection of new functional strains. Other requirements are the absence of potential pathogenic traits (hemolytic, DNase and gelatinase activities) and the study of the antibiotic resistance profile.

It is well known that strains able to transfer resistance to certain antibiotics are of great interest because they can be co-administrated with antibiotics, avoiding side-effects [59,60]. Therefore, in the present study, the *B. asteroides* strains were tested for antimicrobial resistance, following the EFSA guidelines [9]. In accordance with previous studies, the phenotypic approach highlighted that both BA15 and BA17 stains were susceptible to gentamicin, streptomycin, erythromycin, clindamycin and tetracycline [60,61,62,63,64].

Even though the tested *B. asteroides* BA15 and BA17 strains showed phenotypic resistance to ampicillin, vancomycin and chloramphenicol, genome analysis discarded the risk of transferability to the host. Moreover, in the bifidobacterial genome, the *vanZ* genes could be expected to confer low-level resistance to glycopeptide antibiotics, which act by preventing the incorporation of D-Alanine into peptidoglycan precursors. Specific strains of bifidobacteria contain a *vanZ* homolog flanked by a predicted transposase encoding gene (transposon family IS256) [19].

The *Bifidobacterium bifidum* Yakult strain YIT4007 is a mutant of *Bifidobacterium bifidum* Yakult strain YIT4001, showing enhanced resistance to neomycin, erythromycin and streptomycin, due to a chromosomal mutation on genes *rluD* and *rspL*, which increases the resistance to aminoglycosides. In silico analysis has revealed the presence of putative genes for β-lactamase resistance in the *Bifidobacterium* spp. Genome; however, laboratory-based investigations have demonstrated that non-representative strains are resistant to β-lactam antibiotics [65]. Based on collated data from worldwide sources, the European Committee on Antimicrobial Susceptibility testing (EUCAST) [66] software displays the distribution of MIC-values (generated by methods calibrated to broth microdilution or agar dilution) and zone diameters (generated with EUCAST disk diffusion methodology), together with EUCAST epidemiological cut-off values (ECOFFs), and the species *Bifidobacterium asteroides* is not reported among those species subjected to surveillance, such as *B. adolescentis*, *B. angulatum*, *B. animalis*, *B. bifidum*, *B. breve*, *B. catenulatum*, *B. dentium*, *B. longum*, *B. pseudocatenulatum*, *B. pseudolongum*, *B. ruminantium* and *B. thermophilum*. Thus, considering the generic indications for *Bifidobacterium* spp., it is reasonable surveilling the streptomycin and tetracycline resistance for these strains. The range of MIC susceptibility values for these two antibiotics (streptomycin and tetracycline) reported by EUCAST are from 4 to 512 µg/mL and from 0.025 to 512 µg/mL, respectively. These differ from the range values reported by the EFSA (2012) (from 8 to 256 µg/mL for streptomycin and from 2 to 64 µg/mL for tetracycline). According to EUCAST, the highest MIC distribution percentage for streptomycin was 30.77%, with a MIC value of 64 µg/mL and for tetracycline the highest MIC distribution percentage was 25.27%, with a MIC value of 0.5 µg/mL (EUCAST 2019). According to these suggestions, the MIC value of 8 µg/mL shown by the *B. asteroides* BA15 strain for streptomycin is lower than the values reported by EUCAST, for most *Bifidobacterium* spp. (64 µg/mL); in fact, the MIC of the streptomycin was found to be 8 µg/mL in only 3.5% of *Bifidobacterium* spp. The MIC value of 2 µg/mL shown by the strain *B. asteroides* BA15 for tetracycline is higher than that reported by EUCAST for most *Bifidobacterium* spp. (0.5 µg/mL); in fact, the MIC of the tetracycline was found to be 2 µg/mL in only 7.6% of *Bifidobacterium* spp.

The Actinobacteria phylum harbors antibiotic-producing bacteria and carries a large number of resistance genes. In particular, bifidobacteria can harbor resistance genes to macrolide, lincosamide, streptogramin, ketolide, oxazolidinone (MLSKO) and tetracycline, with the following genes: *tet*M, *tet*S, *tet*W, *tet*O, *tet*Q, *tet*L specific for tetracycline, *aph*(E) specific for aminoglycosides *erm*(A), *erm*(X), *erm*(CD) and *erm*(Y) specific for erythromycin [67,68]. Even though the tested *Bifidobacterium asteroides* BA 15 and BA 17 strains showed phenotypic resistance to ampicillin, vancomycin and chloramphenicol, genome analysis discarded the risk of transferability to the host. Genome analysis reveals the absence of antibiotic genes associated to these antibiotics.

The informatic analysis performed on all putative proteins encoded by the *vanZ* gene showed a well-organized conserved domain. In particular, all putative proteins show the presence of the *vanZ* domain. The proteins encoded by *vanZ* 0001 from the *B. asteroides* BA15 genome are the only ones showing two domains, *vanZ* and RDD. The *vanZ* proteins family may be involved in the transport of a still unknown set of ligands because it contains two predicted transmembrane regions; the arginine occurs at the N-terminus of the first helix and the first aspartate occurs in the middle of this helix [69].

The presence of a putative metabolic pathway for vancomycin synthesis in the genomes of both strains is noteworthy. Moreover, in the bifidobacterial genome, the *vanZ* genes could be expected to confer low-level resistance to glycopeptide antibiotics, which acts by preventing the incorporation of D-Alanine into peptidoglycan precursors. Specific strains of bifidobacteria contain a *vanZ* homolog flanked by a predicted transposase encoding gene (transposon family IS256) [19].

The ability to adhere to surfaces, hydrophobicity and auto-aggregation [70] are considered a prerequisite for different applications, such as food for honeybees and antibiotic production. In the present study, in which xylene was chosen as an apolar solvent able to reflect cell surface hydrophobicity [71], the *B. asteroides* BA17 strains exhibited good adhesion to hydrocarbons. In addition, both tested strains showed an auto-aggregation ability similar to that exhibited by the *B. animalis* BB12 reference strain. This feature is essential for epithelium cell colonization preventing elimination by peristalses [72]. Antimicrobial activity against pathogens has been the subject of numerous investigations [59,73,74]. In this study, a broad range of antagonistic activity was displayed by both BA17 and BA15 strains, confirming the ability to inhibit and displace pathogens, as previously reported [75,76]. Several mechanisms have been suggested to explain this inhibitory activity of bifidobacteria towards both Gram-positive and Gram-negative pathogens [74,77,78], such as the decrease in local pH via the production of organic acids, as well as the production of bacteriocins or bacteriocin-like compounds [79,80].

## 5. Conclusions

The present study provides evidence that honeybee gut can be considered a reservoir of bacteria with safety features for human use. This suggests that honeybees could be exploited as an almost unexplored source of isolates for application in different fields, such as food for precious pollinator insects exposed to pesticides and toxic products at sublethal concentrations, and also for antibiotic production.

## Figures and Tables

**Figure 1 microorganisms-10-00655-f001:**
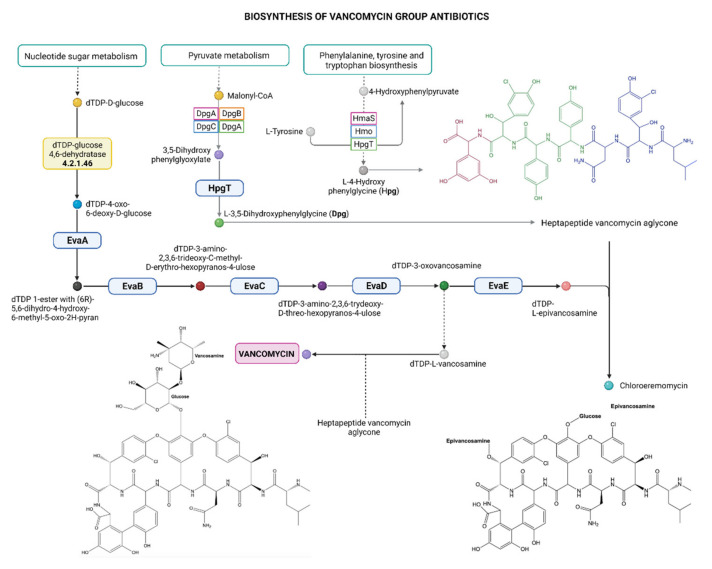
Metabolic pathway responsible for vancomycin biosynthesis. Analysis performed by Patric 3.6.9.

**Figure 2 microorganisms-10-00655-f002:**
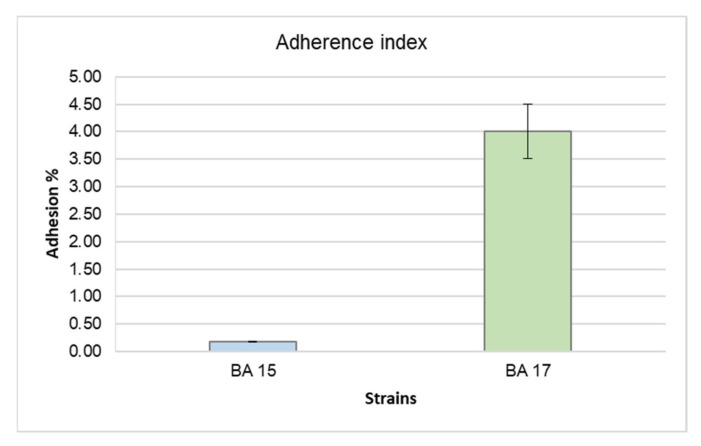
Adherence index of the *Bifidobacterium asteroides* BA15 and *Bifidobacterium asteroides* BA17 strains on an abiotic surface.

**Figure 3 microorganisms-10-00655-f003:**
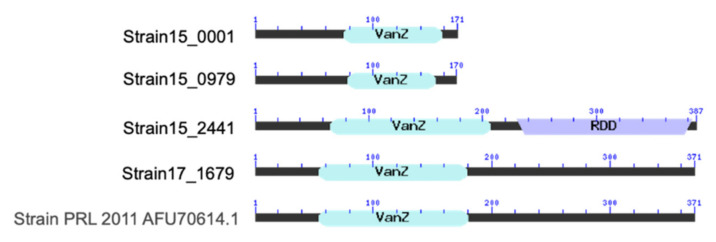
Domain of the putative protein encoded by the *vanZ* gene in the BA15 and BA17 genome, obtained by CDART (domain architectures) from the NCBI database.

**Figure 4 microorganisms-10-00655-f004:**
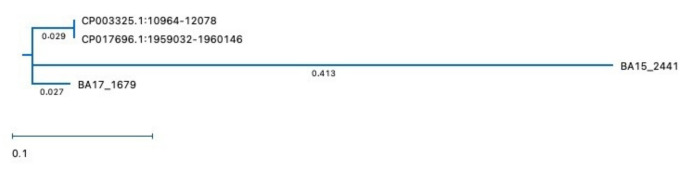
Unrooted phylogenetic tree (constructed by maximum likelihood: RAxML) relating the *vanZ* of BA15 and BA17 and their distance with the DSM 20089 and PRL2011 strains.

**Figure 5 microorganisms-10-00655-f005:**
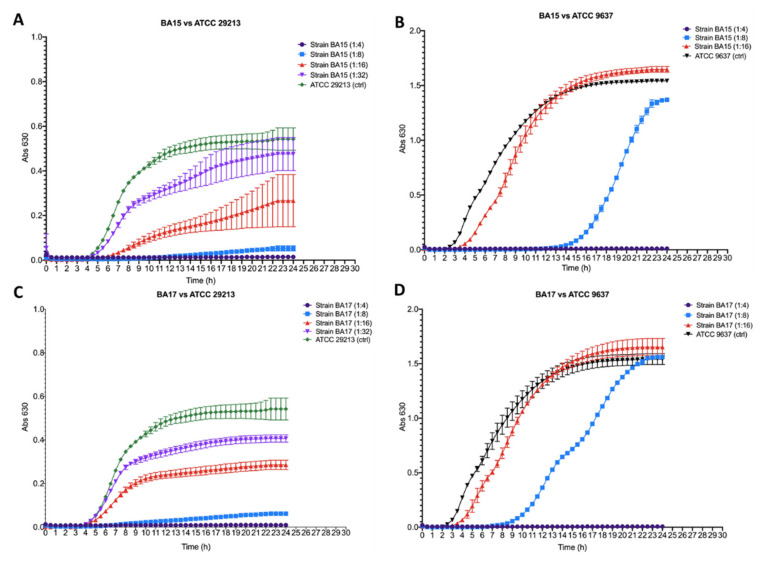
Killing curves of the cells free supernatants (CFS) obtained from 96 h broth cultures of *Bifidobacterium asteroides* BA15 (CFS_BA15_) and *Bifidobacterium asteroides* BA17 (CFS_BA17_) at different dilution factors: 1:4 (purple curve); 1:8 (light blue curve); 1:16 (red curve); 1:32 (light purple curve), against *Escherichia coli* ATCC 9637 (black curve) and *Staphylococcus aureus* ATCC 29213 (green curve). Killing curves of CFS_BA15_ (**A**) against *E. coli* ATCC 9637; (**B**) against *S. aureus* ATCC 29213. Killing curves of CFS_BA17_ (**C**) against *S. aureus* ATCC 29213; (**D**) against *E. coli* ATCC 9637.

**Table 1 microorganisms-10-00655-t001:** PCR primers used for the detection of virulence genes.

Gene	Virulence Factor	Primer Name	Oligonucleotide Sequence (5′ to 3′)	Product Size (bp)
*asa1*	aggregation substance	ASAfw ASArw	GCACGCTATTACGAACTATGA TAAGAAAGAACATCACCACGA	375
*cylA*	cytolysin	CYTfw CYTrw	TATGACAATGCTTTTTGGGAT AGATGCACCCGAAATAATATA	213
*gelE*	gelatinase	GELfw GELrw	ATAGACAATGCTTTTTGGGAT AGATGCACCCGAAATAATATA	213
*hyl*	hyaluronidase	HYLfw HYLrw	ACAGAAGAGCTGCAGGAAATG GACTGACGTCCAAGTTTCCAA	276
*esp*	surface protein	SPfw SPrw	AGATTTCATCTTTGATTCTTGG AATTGATTCTTTAGCATCTGG	688

**Table 2 microorganisms-10-00655-t002:** Enzymatic profile exhibited by the *Bifidobacterium asteroides* BA15 and BA17 strains.

Reaction/Enzyme	BA15	BA17	*Bifidobacterium* spp. (% of Positive Reaction)
Urease	−	−	0
Arginine dehydrolase	+	+	100
α-galactosidase	+	+	100
β-galactosidase	−	−	9
β-galactosidase-6-phosphate	+	+	100
α-glucosidase	+	+	91
β-glucosidase	+	−	45
α-arabinosidase	−	−	0
β-glucuronidase	+	+	64
N-acetyl-β-glucosaminidase	+	+	99
Mannose fermentation	+	+	93
Raffinose fermentation	−	−	0
Glutamic acid decarboxylase	−	−	0
α-fucosidase	−	−	9
Reduction of nitrates	−	−	1
Indole production	−	−	5
Alkaline phosphatase	+	+	100
Arginine arylamidase	+	+	99
Proline arylamidase	−	−	27
Leucyl glycine arylamidase	+	+	99
Phenylalanine arylamidase	+	+	91
Leucine arylamidase	−	−	9
Pyroglutamic acid arylamidase	+	+	99
Tyrosine arylamidase	+	−	64
Alanine arylamidase	+	+	99
Glycine arylamidase	+	+	91
Histidine arylamidase	−	−	1
Glutamyl Glutamic Acid Arylaminidase	+	+	91
Serine arylaminidase	−	−	0

Legend: positive reaction (+); negative reaction (−).

**Table 3 microorganisms-10-00655-t003:** Biochemical profile exhibited by the *Bifidobacterium asteroides* BA15 and BA17 strains.

Biochemical Reactions	BA15	BA17
NH_3_ from arginine	−	−
Gelatin liquefaction	−	−
Indole production	−	−
Glucosidase	+	+
Xylose	−	−
D-Fructose	+	+
D-Galactose	+	+
Maltose	−	−
Trehalose	−	−
D-Melibiose	+	+
Mannitol	−	−
Salicin	−	−
Sorbitol	−	−
L-Arabinose	−	+
Raffinose	−	−
D-Ribose	−	−
Lactose	+	+
Inulin	−	−
Cellobiose	−	−
Melezitose	−	−

Legend: positive reaction (+); negative reaction (−).

**Table 4 microorganisms-10-00655-t004:** Antibiotic resistance pattern of the *Bifidobacterium asteroides* BA15 and BA17 strains.

	AMP (4) *	VAN (2) *	GEN (16) *	STRE (32) *	ERY (1) *	CLI (1) *	TET (8) *	CHL (4) *
	Tested Range (µg/mL)							
STRAINS	(0.5–16)	(0.5–16)	(4–128)	(8–256)	(0.25–8)	(0.25–8)	(2–64)	(1–64)
BB12	<0.5	0.5	128 ^R^	128 ^R^	0.25	<0.25	16 ^R^	2
BA15	16 ^R^	>16 ^R^	4	8	0.25	0.25	2	64 ^R^
BA17	16 ^R^	>16 ^R^	<4	<8	<0.25	<0.25	<2	64 ^R^

*: Microbiological cut-off according to the EFSA Journal, 2008. AMP: ampicillin; GEN: gentamicin; STRE: streptomycin; ERY: erythromycin; CLI: clindamycin; TET: tetracycline; CHL: chloramphenicol; ^R^: resistant.

**Table 5 microorganisms-10-00655-t005:** Antimicrobial activity against pathogenic bacteria.

Strains	*E. coli* *ATCC 25922*	*E. coli* *ATCC 9637*	*S. typhimurium* *ATCC 14028*	*S. aureus* *ATCC 6538*	*S. aureus* *ATCC 29213*
BB12	++	++	++	++	++
BA15	++	+	++	-	+
BA17	++	+	++	++	+

(-) no inhibition zone; (+) inhibition zone <5 mm; (++) inhibition zone >5 mm.

**Table 6 microorganisms-10-00655-t006:** Results of single alignment (Pairwise) between nucleotides of the *vanZ* gene from reference strain PRL2011 (CP003325.1:10964-12078) vs. Strain15_0001, Strain15_0979, Strain15_2441, Strain17_1679.

	Strains	% Identity	% Gaps	Identical	Gap Count	Gap Length	Score	Length
PRL2011	Strain15_0001	46.5	36.9	303	55	240	348	651
	Strain15_0979	49.0	32.8	329	62	220	379	671
	Strain15_2441	49.8	31.5	649	109	409	872	1297
	Strain17_1679	94.0	0.0	1048	0	0	4972	1115

**Table 7 microorganisms-10-00655-t007:** Auto-Aggregation, Co-Aggregation and Hydrophobicity Abilities of the tested strains.

	Auto-A%	Co-A%					H%
Strains		*S. aureus* ATCC 6538	*S. typhimurium* ATCC 14028	*E. coli* ATCC 25922	*S. enteritidis* ATCC 13076	*Listeria monocytogenes* ATCC 12466	
BB 12	36.70 ± 0.11 ^a^	40.05 ± 0.17 ^b^	10.00 ± 0.19 ^a^	23.50 ± 0.13 ^b^	34.50 ± 0.12 ^c^	15.60 ± 0.13 ^a^	84.50 ± 0.13 ^c^
BA 15	34.13 ± 0.13 ^a^	14.22 ± 0.12 ^a^	19.25 ± 0.17 ^b^	15.97 ± 0.12 ^a^	15.97 ± 0.15 ^a^	19.18 ± 0.15 ^b^	59.67 ± 0.14 ^a^
BA 17	33.11 ± 0.17 ^a^	16.67 ± 0.11 ^a^	13.33 ± 0.18 ^a^	28.31 ± 0.18 ^b^	26.25 ± 0.17 ^b^	19.73 ± 0.12 ^b^	79.15 ± 0.11 ^b^

Results are expressed as average value and standard deviation of three separate experiments. The different letters (a–c) in the same column indicate significant differences by one-way ANOVA test, followed by Tukey’s post hoc test (*p* < 0.05). Auto-A%: Auto-aggregation; CoA%: Co-aggregation; H%: Hydrophobicity.

## Data Availability

Not applicable.

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
