# Peer review of "Characterization of *Bifidobacterium asteroides* Isolates"

_microorganisms, 2022, doi:10.3390/microorganisms10030655_

Round 1
Reviewer 1 Report
I think the text is very interesting. the research was well planned. Below are some comments and a question for the authors of the work.
- The Introduction section is long but interesting and introduces well to the subject of the work
- Please write the names of bacteria and genes and ,,in vitro” in italics: See lines: 111 tetW, 316, 491, 492 vanZ 123, 423 in vitro.These are just a few examples, please check the entire manuscript.
- 3.2 section “safety properties” I think that this title should be changed e.g. Biochemical properties, virulence factors and antibiotic resistance profile (lines: 17 and 265)
- in my opinion 2.7.2 section and 3.5 section named Killing activity should be changed e.g. Antibacterial activity or Antibacterial effect
- Unfortunately, Figures 1 and 4 are completely unreadable. Probably because the font is too small or the file resolution is too low.
- Why the authors cite the 2019 EUCAST recommendations, not the newer ones. The recommendation is updated every year.
- Line 454 ,,the European 454 Committee on Antimicrobial Susceptibility testing [66].. Please, add EUCAST acronym: the European Committee on Antimicrobial Susceptibility testing(EUCAST) [66].
- Lines from 465 use mL instead of ml, please check all manuscript.
- Line 472: ,,only 3.5% of Bifidobacterium spp. show a MIC value of 8 µg/ml using streptomycin” – better: MIC of the streptomycin was found 8 µg/mL in only 3.5% of Bifidobacterium spp. MIC is defined as the lowest concentration of an antimicrobial so MIC of vancomycin is a better term.
The following sentence (line 475) requires similar modification ,,in fact only 7.6% of Bifidobacterium spp. show a MIC value of 2 µg/ml 475 using tetracycline.
Author Response
Comments and Suggestions for Authors
I think the text is very interesting. the research was well planned. Below are some comments and a question for the authors of the work.
The Introduction section is long but interesting and introduces well to the subject of the work
Authors: we thank the time devoted by the Reviewer to revise our manuscript. In the current version the English language and style have been improved and minor spell check has been revised.
As suggested by the Academic Editor, the quality of Figure 1-4-5 was improved: the Figure 1 has been simplified to underline the role of dTDP-glucose 4,6-dehydratase (EC number 4.2.1.46) produced by the strains BA15 and BA17, in the biosynthesis of Vancomycin. Thus, font results enhanced, with a better resolution. Figure 4, has been divided: Panel A, whose quality was enhanced, has been maintained within the paper, instead of Panel B, which has been inserted in Supplementary Material category. Figure 5, the color of each growth curve has been standardized, according to the dilution factor, e.g. Dilution 1:4 is always purple; Dilution 1:8 is always light blue, Dilution 1: 16 is always red and Dilution 1:32, if present, is colored in light purple. The reference strain ATCC 29213 is always green colored, whereas the other reference strain, ATCC 9637, is always black colored. In Figure 2 has been added the SD, as required.
Moreover, the bacterial species, have been written in italics throughout the entire manuscript and the references has been standardized with the journal format.
Other requirements have been solved as indicated in the following.
Please write the names of bacteria and genes and ,,in vitro” in italics: See lines: 111 tetW, 316, 491, 492 vanZ 123, 423 in vitro.These are just a few examples, please check the entire manuscript.
Authors: Thanks for the suggestion, we modified the text as you recommended.
3.2 section “safety properties” I think that this title should be changed e.g. Biochemical properties, virulence factors and antibiotic resistance profile (lines: 17 and 265)
Authors: Thanks for the suggestion, the title of the section 3.2 as follow “Antibiotic resistance profile, virulence factors and other biochemical properties” has been modified, according to recommendation and the order of presented data.
in my opinion 2.7.2 section and 3.5 section named Killing activity should be changed e.g. Antibacterial activity or Antibacterial effect
Authors: Thanks for the suggestion, the title of the sections 2.7.2 and 3.5 has been edited, reporting Antibacterial activity.
Unfortunately, Figures 1 and 4 are completely unreadable. Probably because the font is too small or the file resolution is too low.
Authors: Thanks for your comment, the Figure 1 has been simplified to underline the role of dTDP-glucose 4,6-dehydratase (EC number 4.2.1.46) produced by the strains BA15 and BA17, in the biosynthesis of Vancomycin. Thus, font results enhanced, with a better resolution.
Why the authors cite the 2019 EUCAST recommendations, not the newer ones. The recommendation is updated every year.
Authors: Thanks for the comment. We sincerely apologize, the reference has been updated.
Line 454 ,,the European 454 Committee on Antimicrobial Susceptibility testing [66].. Please, add EUCAST acronym: the European Committee on Antimicrobial Susceptibility testing(EUCAST) [66].
Authors: Thanks for the suggestion, the acronym was properly added (please, see line 507).
Lines from 465 use mL instead of ml, please check all manuscript.
Authors: Thanks for your suggestion, the modification from ml to mL has been included in the entire text.
Line 472: ,,only 3.5% of Bifidobacterium spp. show a MIC value of 8 µg/ml using streptomycin” – better: MIC of the streptomycin was found 8 µg/mL in only 3.5% of Bifidobacterium spp. MIC is defined as the lowest concentration of an antimicrobial so MIC of vancomycin is a better term.
The following sentence (line 475) requires similar modification ,,in fact only 7.6% of Bifidobacterium spp. show a MIC value of 2 µg/ml 475 using tetracycline.
Authors: Thanks for your suggestion, the text has been edited as recommended.

Reviewer 2 Report
The authors provided a very interesting study, however it must include some corrections and information before it can be published.
p1 L15-23, p3 L133-134, p16 L517-521. The authors must indicate that the Bifidobacterium strains isolated from the honeybee gut were assessed for safety for human use.
p1 L33-34. The correct name for B. lactis es B. animalis subsp. lactis.
p5 L200-201. Please provide more details about the DNAse and gelatinase activity assays.
p5 L202.204. Hydrophobicity test is frequently performed in many different ways and the reference provided (Pino et al., 2021) does not include any details about it. Was it the BATH test used? If that is the case, please indicate which hydrocarbon was used. This information must be included in the methodology.
p5 L208. What is the importance or reason for doing the test of adhesion to abiotic surfaces? Why did the authors change the culture medium for Bifidobacterium?
p5 L230. Why did the authors used these two strains of E. coli? Any specific characteristics?
p6 L250. Please use "(α=0.05)" instead of "and difference were considered statistically significant at p<0.05"
Author Response
Reviewer #2
Comments and Suggestions for Authors
The authors provided a very interesting study, however it must include some corrections and information before it can be published.
Authors: we really appreciated the time spent and the suggestions of the Reviewer, regarding to our manuscript. The required correction and information have been included.
As suggested by the Academic Editor, the quality of Figure 1-4-5 was improved: the Figure 1 has been simplified to underline the role of dTDP-glucose 4,6-dehydratase (EC number 4.2.1.46) produced by the strains BA15 and BA17, in the biosynthesis of Vancomycin. Thus, font results enhanced, with a better resolution. Figure 4, has been divided: Panel A, whose quality was enhanced, has been maintained within the paper, instead of Panel B, which has been inserted in Supplementary Material category. Figure 5, the color of each growth curve has been standardized, according to the dilution factor, e.g. Dilution 1:4 is always purple; Dilution 1:8 is always light blue, Dilution 1: 16 is always red and Dilution 1:32, if present, is colored in light purple. The reference strain ATCC 29213 is always green colored, whereas the other reference strain, ATCC 9637, is always black colored. In Figure 2 has been added the SD, as required.
The English language and style have been improved and minor spell check has been revised.
Moreover, the bacterial species, have been written in italics throughout the entire manuscript and the references has been standardized with the journal format.
In the following, are punctually reported the modification applied throughout the manuscript.
p1 L15-23, p3 L133-134, p16 L517-521. The authors must indicate that the Bifidobacterium strains isolated from the honeybee gut were assessed for safety for human use.
Authors: Thanks for the suggestion, we modified the text as recommended.
p1 L33-34. The correct name for B. lactis es B. animalis subsp. lactis.
Authors: Thanks for the suggestion, the name for B. lactis has been corrected.
p5 L200-201. Please provide more details about the DNAse and gelatinase activity assays.
Authors: Thanks for the suggestion, more detail about both DNAse and gelatinase activity assays have been added.
p5 L202.204. Hydrophobicity test is frequently performed in many different ways and the reference provided (Pino et al., 2021) does not include any details about it. Was it the BATH test used? If that is the case, please indicate which hydrocarbon was used. This information must be included in the methodology.
Authors: Thanks for the suggestion, more detail about Hydrophobicity test has been reported clarifying that hydrophobicity was determined as bacterial adhesion to hydrocarbons (BATH) through the use of Xylene.
p5 L208. What is the importance or reason for doing the test of adhesion to abiotic surfaces? Why did the authors change the culture medium for Bifidobacterium?
p5 L230. Why did the authors used these two strains of E. coli? Any specific characteristics?
Authors: Thanks for the comment. E. coli ATCC 25922 and E. coli ATCC 9637 are two different target strains. In detail, the E. coli ATCC 25922 strain is a CLSI control strain for antimicrobial susceptibility testing. It is used in several applications, such as antagonistic activity, media testing as negative control, susceptibility disc testing of different antimicrobial agents, food testing, quality control, etc. Moreover, the E. coli ATCC 9637 strain is usually used as control strain for several applications, such as susceptibility testing, pharmaceutical and personal care among others.
p6 L250. Please use "(α=0.05)" instead of "and difference were considered statistically significant at p<0.05"
Authors: Thanks for the suggestion, the modification has been applied on the text as recommended.
